# Barriers and facilitators to HIV and viral hepatitis testing in primary healthcare settings in the Kyrgyz Republic: A mixed-methods study using the COM-B Framework

Ida Sperle[1]*, Nikolay Lunchenkov[1,2], Anastassiya Stepanovich-Falke[1], Zuridin S. Nurmatov[3], Aybek A. Bekbolotov[3], Aisuluu Bolotbaeva[4], Aman Taalaibekov[3], Zamira Abdrakhmanova[3], Zulpueva Akmaral Aidarbekovna[3,5], Michael Brandl[1], Olena Kysil[1], Stela Bivol[6], Viviane Bremer[1], Barbara Gunsenheimer-Bartmeyer[1☯], Sandra Dudareva[1,7☯]

1 Robert Koch Institute, Department of Infectious Disease Epidemiology, Berlin, Germany, 2 Technical University of Munich, TUM School of Social Science and Technology, Munich, Germany, 3 National Institute of Public Health of the Ministry of Health of the Kyrgyz Republic, Bishkek, Kyrgyz Republic, 4 Independent consultant, Bishkek, Kyrgyz Republic, 5 Osh State University, Osh, Kyrgyz Republic, 6 WHO Regional Office for Europe, Copenhagen, Denmark, 7 Institute of Public Health, Riga Stradins University, Riga, Latvia

☯ Barbara Gunsenheimer-Bartmeyer and Sandra Dudareva shared last authorship
* sperle-heupeli@rki.de

## Abstract

### Introduction

The Kyrgyz programme on the elimination of HIV and ending AIDS and viral hepatitis infections (2023–2027) recognizes the need to scale up testing and include primary healthcare. Our aim was to identify and analyse important barriers and facilitators to HIV and hepatitis B, C and D testing from the perspective of medical doctors working in primary healthcare in the Kyrgyz Republic.

### Methods

We conducted a cross-sectional, mixed-methods study in the Kyrgyz Republic from June to November 2024, and first conducted in-depth semi-structured interviews. We applied a thematic analysis for qualitative analysis. We developed a questionnaire based on these main themes, which was distributed to medical doctors working in primary healthcare. A descriptive analysis of the questionnaire data followed.

### Results

Through 21 interviews, we identified training gaps, lack of knowledge, lack of time and physical space as well as social and cultural perceptions as key barriers. Data from 1,080 questionnaire responses (89% female; 46% 61 years or older) supported

**Data availability statement:** The datasets described in this article are not publicly available to protect the privacy of interviewees and questionnaire participants. Despite anonymizing directly identifiable information within the interview transcripts and questionnaire data, the risk of indirect identification persists, owing to detailed descriptions of organizational roles and geographic locations. Reasonable requests to access parts of the dataset (excluding sections with potential identifiers) can be directed to the Research Data Center of the Robert Koch Institute (fdz@rki.de).

**Funding:** This study was funded by the Federal Government of Germany under the framework of the Global Health Protection Programme.

**Competing interests:** The authors have declared that no competing interests exist.

this with 35% reporting lack of time, 39% lack of doctors, and over 60% indicating need for further training. Identified facilitators included knowledge of guidelines, new technologies, e.g., self-testing, and the possibility of linking testing to hepatitis B vaccination. Questionnaire data confirmed access to free testing and treatment (HIV: 86%, viral hepatitis 89%), knowledge of testing indications (HIV: 80%, viral hepatitis: 76%), and linkage to vaccination (71%) as important facilitators. Lack of experience with key populations was evident with 78% not identifying any members of key populations as patients.

## Conclusion

To increase testing and reach the goals of ending AIDS and the epidemics of viral hepatitis, doctors in primary healthcare need capacity-building in the form of HIV and viral hepatitis training and knowledge dissemination to offer testing. Moreover, the availability and distribution of doctors in primary healthcare need to be prioritized and enhanced in areas in most need of care to ensure access to testing for all in the Kyrgyz Republic.

## Introduction

In the World Health Organization (WHO) European Region, an estimated 3 million people are living with HIV, and 10.6 million and 8.6 million people with hepatitis B and C, respectively [1]. The burden is particularly high in the Eastern European and Central Asian countries [2], where the HIV and viral hepatitis incidence continues to increase [3]. The WHO regional action plans for ending AIDS and the epidemics of viral hepatitis and sexually transmitted infections (STIs) (2022–2030) [4] outline specific goals and actions needed to reach elimination. The action plans underline the importance of a people-centred approach, and integration and decentralization of services including testing for HIV and viral hepatitis (and other STIs) in primary healthcare (PHC) settings. Better access to free testing and treatment is needed. In the WHO European Region, it is estimated that 23% of people living with HIV are unaware of their HIV infection, while only 16% of people with chronic hepatitis B and 29% with hepatitis C have been diagnosed [1]. The targets for testing to be met by 2030 to end AIDS and the epidemics of viral hepatitis are defined as 90% of those living with hepatitis B or C should be diagnosed, and 95% of people living with HIV should know their HIV status.

In the Kyrgyz Republic, the estimated prevalence of hepatitis B virus (HBV) (presence of hepatitis B surface antigen (HBsAg+) and hepatitis C virus (HCV) (chronic HCV (RNA+/cAg) is 5.3% and 2.5%, respectively [5,6]. The prevalence of HIV, estimated in 2021, was 0.3%, with key populations being the most affected groups [3]. The estimated HIV prevalence among key populations (2023) was 18% and 11% among people who inject drugs and among men who have sex with men, respectively [7]. In the Kyrgyz Republic, it was estimated that 79% of people living with HIV

are aware of their HIV infection [7], while for HBV and HCV this number remains unknown. However, for HBV and HCV, before 2023, free testing was available mainly to people living with HIV, as part of epidemiological investigations or in case of suspicion of acute infection. It is therefore assumed that testing uptake and number of people aware of their infection is low in the general population and key populations. The Ministry of Health (MoH) and Social Development is responsible for the healthcare system in the Kyrgyz Republic. Healthcare service is funded through the Mandatory Health Insurance Fund, and most healthcare facilities are public. However, services covered are still somewhat limited, and some care still requires co-payments which remains a barrier to many people. Moreover, public spending on health is limited, and some people are still not enrolled in the mandatory health insurance scheme [8,9].

The national programme on the elimination of HIV and viral hepatitis in the Kyrgyz Republic (2023−2027) [10] recognizes the need to scale up testing to close the gap in diagnoses in the population. The programme specifically mentions for example blood donors, healthcare workers, pregnant women, and other groups at higher risk of infection such as key populations, people living with HIV (for viral hepatitis) and people with sexually transmitted infections (STIs) and other patients with chronic conditions. HIV testing and access to effective antiretroviral therapy (ART) is publicly funded, and available at PHC level in both urban and rural areas. HIV testing and treatment is also available at state funded regional AIDS centres. Nonetheless, there is an uneven distribution of healthcare workers, and there are more medical doctors, also on PHC level, in urban areas compared to rural areas. This skewed distribution was further enhanced during the COVID-19 pandemic [11]. HBV and HCV testing and treatment has been made available and free of charge through PHC since 2023. An important part of the programme and ensuring access to, and increase of, testing is the decentralisation of testing, of which the allocation of HIV and viral hepatitis testing to PHC is a key part. To ensure quality and infectious disease expertise in PHC, each PHC setting should also have one infectious disease doctor who is responsible for patients testing positive.

Late diagnosis of HIV and viral hepatitis has large health consequences for the individual but also public health. For viral hepatitis, there is a need to improve timely testing in the Kyrgyz Republic [12] to link to care and prevent late sequelae such as liver cirrhosis and hepatocellular carcinoma. For HIV, timely and regular testing is important to ensure diagnosis and prevent late HIV diagnosis which increases mortality and morbidity, and risk of onward transmission. In the Kyrgyz Republic, nosocomial transmission of viral hepatitis (in particular) and HIV remains a concern due to inadequate infection control practices in some hospitals and healthcare facilities making it a common risk factor for the general population.

Due to the most common routes of transmission of HIV and viral hepatitis, certain sub-populations are at increased risk of infection, and co-infections are common warranting integrated testing [13]. Moreover, offering HIV and viral hepatitis testing in PHC settings through decentralisation and task-shifting from specialty healthcare has been proven both feasible and acceptable [14–17]. PHC is also uniquely positioned as the first contact point in the healthcare system to offer people who do not seek specialised care, HIV and viral hepatitis testing [8]. However, it is imperative that PHC is equipped with the necessary knowledge and resources, including securing availability of tests, including rapid tests.

People at a higher risk of infection with HIV or viral hepatitis, including key populations, often face different social, legal and structural barriers in accessing PHC services. Therefore, many key populations seek care through community-based facilities with programmes specific for key populations. Many are led by non-governmental organisations and funded through international donors. For PHC to successfully implement testing also for key populations, these barriers need to be addressed to ensure equitable and safe access to HIV and viral hepatitis testing [18]. Uncovering factors and processes that influence testing in PHC can be structured by identifying barriers, defined as internal or external factors that either prevent or complicate testing, and facilitators defined as internal or external factors that helps or encourage testing.

The overall aim of the study was to outline the main barriers and facilitators for doctors working in PHC to test for HIV and hepatitis B, C, and D in the Kyrgyz Republic. More specifically, our objectives were, first, to identify and describe key

barriers and facilitators for HIV and viral hepatitis testing perceived by doctors working in primary PHC using the COM-B framework (capability, opportunity and motivation). Second, to describe key characteristics of doctors working in PHC focusing on HBV, HCV and HIV testing activities and experience with healthcare for key populations. Third, to analyse the significant differences between reported barriers and socio-demographic information.

## Materials and methods

We have published the detailed methodology as well as challenges in conducting this study elsewhere [19], and describe the methods in brief below.

We conducted a cross-sectional mixed methods study, which was carried out in two phases. We used semi-structured in-depth interviews (IDIs) (qualitative part, phase I) and an online questionnaire (quantitative part, phase II) to explore barriers and facilitators for doctors working in PHC to test for HIV and hepatitis B, C and D in the Kyrgyz Republic. The qualitative part of the manuscript has been reported in line with Consolidated Criteria for Reporting Qualitative Research [20].

### Study team

The research team was composed of one research team based at the Robert Koch Institute (RKI) in Germany and one research team in the Kyrgyz Republic. The research team in the Kyrgyz Republic comprised of national partners from the local public health authorities such as the National Institute of Public Health of the MoH and the Republican Center for Blood-borne Viral Hepatitis and HIV/AIDS Control at the MoH.

The research team in the Kyrgyz Republic coordinated in-country activities including the organisation of the IDIs and the distribution of the online questionnaire through regional health coordinators, who then contacted the PHC facilities. Regional health coordinators are appointed medical doctors responsible for overseeing HIV and viral hepatitis work in remote areas.

### Study population

We included medical doctors currently working in PHC regardless of professional background for both phase I and II. We included PHC doctors working in the regions Bishkek (the capital city), Chuy Oblast, Issyk-Kul Oblast, Talas Oblast, Osh Oblast (including Osh City), and Manas Oblast, representing both urban and rural areas with different population densities and levels of primary health infrastructure.

### Sampling and recruitment

**Phase I: qualitative part.** We recruited participants through the research team in the Kyrgyz Republic. We used a multi-stage approach to select the PHC facilities. By using a list of PHC facilities provided by the Kyrgyz research team, facilities were selected based on geographical location, testing volume (2023) and population density. Local civil society organisations working with people with HIV and viral hepatitis and key populations were consulted on reported experienced stigma and discrimination within certain health facilities for contextual information. The details of the selection process are published elsewhere [19].

Prior to participation, the doctors in PHC received a detailed study information sheet in their preferred language (Kyrgyz or Russian), and provided written informed consent. Participants could opt to use pseudonyms to protect their identity. Each participant received the equivalent of 20 Euro in local currency as compensation for their time.

Our target sample size was 18–20 participants, which is in line with recommendations for medium-sized qualitative studies [21,22]. In order to assess repeated response patterns (data saturation) on a running basis, the research team conducted daily debriefing sessions.

**Phase II: quantitative part.** An official letter informing about the study and the online quantitative questionnaire was prepared by the Kyrgyz research team, and sent out on behalf of the deputy minister of the MoH to all regional health

coordinators in the Kyrgyz Republic. The letter with a link to the online questionnaire was then forwarded to doctors in PHC on a district and city level via WhatsApp groups and E-Mails. The participants responded anonymously, and no incentives were provided for participation in the quantitative questionnaire. The questionnaire remained open from 04 to 29 November 2024. The target sample size was set to 400 participants. Updates on the number of participants, who had completed the questionnaire as well as the region in which their PHC facility was located, were provided by the RKI research team to the Kyrgyz research team on a weekly basis. This enabled targeted follow-up by the Kyrgyz research team with PHC facilities and/or geographical areas with low participation.

## Data collection

**Phase I: qualitative part.** We developed a semi-structured interview guide based on an extensive literature review examining barriers and facilitators to HIV and viral hepatitis testing in PHC settings, with particular attention to studies from Central Asia and similar resource-constrained settings [23–25]. A series of consultations were held with local stakeholders in the Kyrgyz Republic, including public health experts and civil society representatives, who provided feedback that informed the content of the guide. The draft underwent several rounds of revision based on feedback from academic experts both in the Kyrgyz Republic and Germany, as well as local health professionals. The guide was tested within the RKI research team, and piloted with participants who were not part of the final study sample. Lessons learnt from the pilot were incorporated into the final version, which was finalised in English and translated into Kyrgyz and Russian by a certified translator [see S1 Appendix].

The interview guide explored doctors in PHC perceptions of barriers and facilitators to HIV and viral hepatitis testing, with a particular focus on current testing practices and knowledge. Data collection took place from 19 to 26 June 2024, and the interviews were conducted by three researchers (ASF, NL and AB) with expertise in public health and qualitative research methodology. Interviews were conducted in the language preferred by the participants (Russian or Kyrgyz). Each interview took place in a private room at the participant's PHC facility and lasted approximately 60–90 minutes, with only the interviewer and participant present to maintain confidentiality. With participants' informed consent, all interviews were audio-recorded and then transcribed verbatim for analysis by two members of the Kyrgyz research team. Personal information, the anonymised recordings and transcriptions were saved on a separate secured drive. Recording-files and transcripts were transferred using an encrypted exchange server (Cryptshare). Recordings were deleted after transcription. All data transfers and storage procedures adhered to RKI data protection guidelines. The translated interviews were proof-read, quality-checked and corrected if needed by the researcher who had conducted the interview. The researchers doing the quality checks were native speakers of Russian or Kyrgyz and checked quality by comparing the English translations with the original transcripts. This linguistic verification process helped to render the authenticity of participants' expressions and contextual meanings throughout the data analysis. In addition, logistical constraints prevented us from sharing the final transcripts with study participants.

Interviewers documented observations through postscript reflections to capture contextual elements beyond the audio recordings. These reflections captured non-verbal communication, emotional responses and situational dynamics as well as immediate observations on key themes and preliminary analytical insights that emerged during the interview. These postscripts were not analysed in addition to the transcripts, but they aided data triangulation by providing an additional perspective beyond the verbal transcript.

**Phase II: quantitative part.** We developed an electronic quantitative questionnaire [see S2 Appendix]. The questionnaire included 30 questions and covered sociodemographic information, HBV, HCV and HIV testing activities, self-reported hepatitis B vaccination status, barriers and facilitators and key populations. We used a likert-scale ranging from strongly disagree to strongly agree for the recurrent barriers and facilitators identified in the IDIs. The questions covering stigma, discrimination and key populations were adapted from a survey developed by the European Centre for Disease Prevention and Control (ECDC) and European AIDS Clinical Society [26]. The questionnaire was reviewed by the Kyrgyz research team, and the final version was translated into Kyrgyz and Russian.

The questionnaire was set up in an online survey tool (Voxco), and the landing page included detailed information about the study. Prior to proceeding to the questions, the participants needed to confirm working in PHC, being minimum 18 years of age and provide their informed consent to participate in the study.

## Data analysis

**Phase I: qualitative part.** Two researchers (IS and NL) analysed the qualitative data following Braun and Clarke's six-step process for thematic analysis [27]. In the first step, both researchers independently familiarised themselves with the interview transcripts after which IS and NL proceeded with preliminary coding using the first three interview transcripts. Both researchers coded these transcripts independently, developing a wide range of potential codes without the influence of each other's interpretations and backgrounds. Once the initial coding was completed, the two researchers had several meetings to refine the initial codes, merge overlapping concepts and develop a consolidated coding framework. This framework was then applied to the remaining transcripts proceeding to a more structured deductive approach, while remaining open to developing new codes. Then, in the third step IS and NL began to develop a preliminary thematic map outlining the main themes and sub-themes developed from the data. After identifying potential themes, in step four, IS and NL shared their codes and themes with the RKI research team for discussion and refinement. As a result of these discussions, some themes were modified, merged or discarded as irrelevant to the research question, while others were further refined. The final coding scheme can be found in S3 Appendix. Finally, the codes were organised into three main themes for each section, barriers and facilitators, reflecting the Capability, Opportunity and Motivation (COM-B) framework [28]. This framework offers a flexible approach to understanding behaviour, here testing, in particular regarding barriers and facilitators structured according to capability, opportunity and motivation for a certain type of behaviour in any context. Thus, the main three themes for each section were COB: Capability, Opportunity and Motivation for HIV and viral hepatitis testing. The final step involved synthesizing these themes into a narrative that addressed our research question.

**Phase II: quantitative part.** We performed descriptive analyses in R 4.2.2 (with the packages 'dplyr' and 'epitools'). We derived frequencies and proportions of sociodemographic information. We excluded participants who reported that their main place of work was outside PHC or polyclinics.

For the geographical location of PHC facilities, these were grouped in Northern and Southern part of the country, with Batken, Manas, Osh Oblast (including Osh City) belonging to the South, and Bishkek (the capital city), Chuy Oblast, Issyk-Kul Oblast and Talas Oblast to the North. Age was grouped into two categories for the stratified analysis (18–50 and over 51 years of age).

We derived frequencies for each of the likert-scale categories for all the barriers and facilitators. We examined the perceived barriers and facilitators stratified by age and geographical location, and we used Pearson's Chi-squared test to look into whether there were any significant differences ($p < 0.05$) between the most frequently reported barriers and age and geographical location.

## Author reflexivity and positionality in analysis of qualitative data

For the qualitative part of this study, we provide a detailed analysis of reflexivity, focusing on how our professional and personal experiences might have influenced our methodological approach and interpretation of findings.

**The first author.** The first author (IS) has a public health background, and is specialised in infectious disease epidemiology. She has experience with viral hepatitis and HIV, as well as key populations. Her previous experience with key populations might have influenced a particular focus on and interest for these groups. As a female coming from a Western European country this may have impacted the way the data were analysed and interpreted.

**The second author.** The second author (NL) brings a dual perspective to this research as both a medical professional with clinical experience in HIV medicine and as a gay man from Eastern Europe with cultural ties to Central Asia. His roles informed the research, combining clinical expertise with personal insights to better identify discriminatory practices

and barriers, particularly around HIV testing. This dual perspective proved valuable in the interviews, fostering a 'doctor-to-doctor' interaction with health workers while ensuring scientific rigour. At the same time, his background required awareness of potential biases.

### Ethics approval and consent to participate

This study was approved by the ethics committee of the institutional review board, journal of health care, in the Kyrgyz Republic, under approval number 01–3115. All study participants, in both the qualitative and quantitative part, provided written informed consent prior to participation.

## Results

### Part I: qualitative results

This section presents the findings from our qualitative analysis of barriers and facilitators to HIV and viral hepatitis testing in PHC in the Kyrgyz Republic. The analysis is structured according to the COM-B framework, examining how capability, opportunity, and motivation factors influence testing practices among doctors in PHC.

### Barriers to HIV and viral hepatitis testing

***Capability: Heterogeneous distribution of knowledge and training gaps.*** Doctors in PHC demonstrated varying levels of competence in HIV and viral hepatitis testing in PHC settings. While participants demonstrated adequate knowledge of clinical indicators for testing, including signs and presenting symptoms for both HIV and viral hepatitis, they described significant challenges in recognising epidemiological criteria, particularly those related to sexual transmission and substance use behaviours that go beyond a number of sexual partners and condom use. Doctors mentioned that they were able to list the clinical indications for HIV testing better than for viral hepatitis, because national protocols for viral hepatitis do not provide similar clear guidance on indications for testing. They also highlighted important gaps in knowledge that may exist among their colleagues about the practical aspects of testing, including the proper provision of testing and follow-up of positive results. As one doctor noted:

> In the last six months, we haven't had any training on HIV, AIDS or viral hepatitis. The training sessions have covered other topics. (P4)

Notably, participants reported that their knowledge of HIV testing came mainly from specialised training provided by organisations such as [the Republican AIDS Centre] or [ICAP], rather than from standard medical education or predominant in-house training. This reliance on external training mechanisms contributed to an uneven distribution of knowledge among health care providers, as one doctor explained:

> "...my knowledge is mostly gained through training I attended; I can't say that everyone [...] is well-informed. (P4)

Health workers also pointed to a clear disparity in training opportunities between HIV and other non-communicable diseases:

> I would like them [Ministry of Health] to offer additional courses [...]. We have a lot of training courses on diabetes and hypertension, and nothing about HIV. (P18)

In addition to HIV-specific training gaps, participants described inadequate training in viral hepatitis testing protocols, interpretation of results and approaches to patient counselling.

***Capability: Clinical (in)experience and population (in)visibility.*** Clinicians reported minimal exposure to and experience with both people living with HIV and key populations in their primary care practice. Some participants reported a complete lack of encounters with people living with HIV and key populations in their clinical practice throughout their careers. As one participant put it:

*Honestly, I have never had an HIV-positive case in my practice.* (P2)

This limited clinical exposure was particularly pronounced in relation to key populations, with doctors often expressing doubt about the very existence of such groups in their practice setting or area of residence. When discussing sex workers, people who inject drugs and men who have sex with men, participants often suggested that these populations were absent from their communities, either because it was difficult for them to exist due to high levels of stigma and potential exposure to violence, or because of *'high moral standards'* in the communities. One participant noted:

*I have never seen such patients [key populations] in outpatient clinics.* (P3)

while another, working in a rural setting, remarked:

*No, we don't have sex workers here. Where would you find sex workers in the village?* (P15)

While participants in all regions described limited contact with key populations, doctors from southern regions of the Kyrgyz Republic had very different attitudes and experiences than their northern counterparts. Thus, we observed among northern participants more open and less judgmental attitudes among doctors. Doctors mentioned challenges in identifying key populations among their patients, noting that discussions about sexual behaviour or substance use were uncommon in their practice, or that they would feel extremely uncomfortable doing so with their patients. Some participants also described having limited information on how to work with key populations when encountering them in their healthcare facility.

***Opportunity – Physical: Time, space and resource.*** Our study participants described substantial resource constraints that limited their ability to provide HIV and viral hepatitis testing services in PHC settings. These constraints manifested themselves in several areas of healthcare delivery, from basic infrastructure such as the laboratories themselves to specialised testing equipment. Some of the health facilities, particularly those in remote areas, faced significant challenges with laboratory infrastructure, with some of them reporting operating without basic laboratory facilities. One study participant practising in one of the regions highlighted the deterioration of laboratory infrastructure over time:

*We are the only district that doesn't have a laboratory. We used to have one but now we don't have one.* (P12)

Doctors in PHC described limitations in test supplies, particularly for viral hepatitis testing. The lack of comprehensive test supplies emerged as a significant barrier, with one participant noting:

*We've never had tests for Hepatitis D. As for Hepatitis C, they [local lab facility] recently told us they couldn't perform quantitative PCR anymore, only qualitative PCR tests. Regarding Hepatitis B, they also stopped doing quantitative PCR….* (P20)

Staff shortages emerged as another critical barrier, with doctors in PHC reporting significant understaffing in their facilities. This shortage affected not only testing services, but the overall capacity of health facilities to provide care to their patients. As another study participant described the situation:

*…we are talking about the primary care level, where it is expected that 8 family physicians should work, there are only two physicians working, or there are 2-3 villages to be covered, and there is only one physician available for a population of 5-6 thousand people, it could be even more.* (P19)

These resource constraints were particularly vocal in remote health facilities, where doctors in PHC described working with minimal staff while trying to maintain testing services alongside other primary care responsibilities such as diagnostics, preventive care, vaccination, and chronic non-communicable disease management. Doctors repeatedly emphasized challenges in maintaining adequate supplies of test materials and ensuring proper test storage conditions, particularly in facilities with unreliable electricity or lack of proper refrigeration:

*…there are power outages, and the Internet sometimes goes down too. It happens quite often. We have generators at every PHC facility to support power supply. But if the Internet goes out, we just have to wait.* (P17)

At the same time, several participants highlighted the lack of private space for individual patient counselling as a structural barrier in some healthcare facilities. Doctors reported that they often had to share consulting rooms with other doctors or medical staff, which limited their ability to have confidential discussions. This was seen as a significant barrier for both providers and users – particularly when discussing sensitive issues such as sexual behaviour or substance use, where privacy is essential:

*Two physicians and three nurses share the office, and when patients visit each physician at the same time, there are 8 people in one room, and if they have children, there are even more, it's like a bazaar.* (P10)

Beyond the resource constraints, time and geographical barriers were perceived by participants to make access to testing services more difficult. The organisation of service delivery, particularly in terms of opening hours, emerged as a recurring narrative and a challenge for the working population. Doctors in PHC acknowledged that standard clinic hours often coincided with typical working hours of the general population complicating access to testing. Doctors in PHC also described geographical barriers to testing services, particularly in remote areas where patients sometimes had to travel long distances to reach testing facilities.

***Opportunity – Social: Social perceptions, culture and their role in testing uptake.*** Study participants described how societal stigma and discrimination as well as religious and cultural norms created barriers for people seeking HIV and viral hepatitis testing services. They described how patients often reacted defensively to testing offers, as doctors in PHC perceived, mainly due to fear of being associated with key populations. This fear of association often manifested itself in immediate defensive reactions from patients, as one doctor told us:

*When they hear HIV, they immediately associate it with prostitutes, people who use drugs and say, 'who do you think I am?'* (P15)

Religious beliefs and cultural expectations appeared to be intertwined with social perceptions, shaping not only how testing was viewed, but also how conversations about testing evolved. These norms often influenced patients' (dis)comfort in discussing sensitive topics such as sexual behaviour and often dictated their openness to testing. As one doctor in PHC told us:

*We have patients who are religious, for example. They have difficulty with answering questions. It is difficult to ask them about such a topic as family relations.* (P18)

The influence of religious beliefs on health-seeking behaviour was particularly evident in certain areas, where healthcare workers noted varying levels of resistance to testing services. Some patients demonstrated a complete refusal of medical intervention, relying on divine protection. One study participant told us:

*I also have very religious patients who just say: 'God will protect me' and they deny any treatment.* (P4)

Doctors in PHC described specific challenges in working with religious families, particularly in relation to gender dynamics and spousal testing. These challenges were felt most acutely in the context of antenatal care, where doctors struggled to engage male partners in testing services. The interplay between religious beliefs and gender norms created additional barriers to testing uptake, particularly when trying to involve both partners in health services:

*We are Muslims, yes. Our religious provisions. When we examine pregnant women, we tell their husbands to get tested. There are cases in which they refuse. There are men like that.* (P7)

**Motivation: Moral judgements and professional conflicts.** While doctors in PHC recognised stigma, cultural and religious norms as socially-caused barriers to testing, they often held stigmatising attitudes themselves during interviews without self-awareness, particularly in relation to perceived (im)moral behaviour and patterns of disease transmission in their communities. Some participants attributed rising infection rates to what they perceived as a decline in social and moral values and changes in sexual behaviour, such as an increase in the number of sexual partners. These perceptions were often framed within a broader narrative of moral decline:

*Our generations have also changed in a bad way. Sexual promiscuity, that is why HIV is growing, and hepatitis is growing somehow.* (P7)

The complexity of stigma was further illustrated by the seemingly contradictory attitudes of doctors in PHC towards key populations. While some doctors dismissed the existence of key populations in their service areas (*see 'Clinical (in)experience and population (in)visibility'*), they also made moral judgements about transmission routes, often focusing on sexual transmission and potentially downplaying other routes of infection, which may have minimised their motivation to test:

*We don't have drug addicts, for God's sake. So, the main transmission is sexual.* (P15)

**Motivation: Patient misconceptions, misbeliefs and indifference about testing practice.** Finally, doctors described their perceptions of strong misconceptions about testing and disease risk that negatively affect their patients' motivation to be tested. According to the participants, many patients underestimate the seriousness of HIV and viral hepatitis and do not recognise the importance of early testing and prevention:

*They [patients] don't have this information and do little self-education regarding medical care. They don't think about the possible consequences for themselves. They don't come for examination until they get concerned about their complaints.* (P18)

Another recurring theme in the interviews was peoples' lack of understanding of their personal responsibility for their health. They believed that this attitude manifested itself in indifference or denial of potential health risks, creating self-imposed barriers to testing. Doctors described encountering consistent patterns of avoidance and procrastination, with people prioritising other activities over their health needs, or waiting until symptoms became severe before seeking care:

*I would say that we feel more responsibility than the patients. They act like 'if you'd invited me, if you'd explained it to me, if you'd told me, then I would come'. They don't have that level of responsibility, and they don't think like: 'It's time to get tested, I have to do it now'.* (P16)

This lack of proactive health-seeking behaviour was seen by doctors as an important barrier to early testing.

### Facilitators to HIV and viral hepatitis testing

***Capability: Increased awareness through public health communication.*** While doctors noted persistent misconceptions and indifference among some patients about testing (see "*Patient misconceptions, misbeliefs and indifference about testing practice*"), they also observed a gradual improvement in overall public awareness of HIV and viral hepatitis testing. This progress was attributed to increased media coverage and improved access to health information among other things enabled by digital platforms and social media. This shift in information accessibility was perceived to facilitate testing uptake, as described by one participant:

*Those who come to me, they are all tested, because the information is now given through the TV, the media, they are now fully informed.* (P3)

Not surprisingly, doctors perceived that the transition to digital health communication appeared to be particularly effective among younger populations, who demonstrated more proactive health-seeking behaviour. Doctors in PHC noticed that some patients were increasingly arriving at facilities with prior knowledge of testing, albeit with varying levels of understanding the difference between HIV and viral hepatitis. This evolution in patient knowledge and engagement was seen as a positive change, with doctors noting a marked change in patient-provider interactions around HIV and viral hepatitis testing:

*Younger people are generally well-informed because they're online a lot. They look things up immediately and sometimes even correct me.* (P4)

According to participants, this increased public awareness through multiple media channels reduced stigma around testing and created opportunities for people and their patients to be better informed about prevention, testing and treatment options.

***Capability: Improving professional awareness and knowledge.*** Interviewees demonstrated a deep understanding of the public health implications of HIV and viral hepatitis in primary care settings, and described knowledge-sharing mechanisms within their organisations to enable them to improve the response to HIV and viral hepatitis. First, doctors in PHC articulated a clear recognition of the impact of those diseases on the health and well-being of the population, identifying both short and longer-term public health effects. This awareness was particularly pronounced in relation to the age distribution of the populations affected and the potential progression to serious health consequences if infections are left undiagnosed or untreated:

*…they are chronic infections, chronic diseases, in my experience generally they occur in the working age from 20 to 40, 40 to 50 years of age. They have a detrimental effect on health and even this leads to disability and with subsequent mortality if it is detected at a late stage.* (P14)

Second, they described how health facilities had established regular information-sharing mechanisms to ensure that doctors were kept up to date with new guidelines and protocols. As described by participants, once a new regulation is

issued by the MoH, it is published on the MoH website shortly afterwards, and the regulation is also emailed to primary care facilities and then to all doctors.

*Our administration gives them to us. There is a meeting once a week to discuss it. It is given to the whole department. The new order comes to us immediately.* (P3)

In order to ensure that the document is not just widely circulated, the participants told us that they have regular meetings where these regulations are discussed. The doctors saw this mechanism not only as a quick way of disseminating knowledge and information about the new regulations, but also as a useful approach to raising awareness in remote areas and thus improving uptake of testing.

***Opportunity – Physical: Technology and outreach solutions.*** Doctors described several successful initiatives and technological advances that have transformed their ability to deliver testing services, particularly in overcoming traditional barriers to access. The use of mobile testing units emerged as a particularly effective solution for reaching underserved populations and remote communities. These mobile clinics, equipped with testing facilities, brought comprehensive services directly to where people live and work, including remote areas of Kyrgyzstan. Our study participants emphasised that this approach removed transport barriers and saved time for patients:

*Yes, we work with schools, and we have voluntary blood donation campaigns... It's convenient so they don't have to come here. They also test there on the spot and draw blood samples for the testing. It is convenient for local communities' dwellers not to come here, not to spend extra money, not to spend extra time. This is what we have well organised, and it has a good effect. (P19)*

This expanded access was further enhanced by the introduction of rapid testing technologies, which were consistently identified by doctors in PHC as a key facilitator for more efficient and accessible services. The shift from traditional laboratory-based testing to point-of-care diagnostics was an important improvement, specifically, in HIV service delivery, as noted by another doctor:

*With the introduction of rapid tests, it has become better, people started coming to us... Before the rapid tests we had ELISA for this testing. It took about a week to get the result, but that rapid test is very sensitive, it gives the result in 20 minutes, and we can refer this patient to a specialist for consultation and register the patient and take him to treatment from the first detection to reduce the time. (P14)*

Doctors in PHC felt that this synergy between mobile outreach services and rapid testing technologies had created a more responsive and accessible testing infrastructure, particularly benefiting communities that traditionally faced barriers to facility-based services, while increasing patient engagement and follow-up opportunities.

***Opportunity – Social: Adapting to reality and building bridges to testing services.*** A major shift in health policy and service delivery emerged as a key facilitator with the introduction of free testing and treatment services. Doctors in PHC consistently highlighted how the removal of financial barriers had changed service delivery and peoples' willingness to engage with testing services. As one provider noted:

*Previously, as I said, until May last year, these tests were paid for, about 260 soms [approximately 3 Euro] according to the price list, now these tests are free... Well, I think the fact that we have free treatment for HIV and hepatitis, the fact that we have free treatment and testing, that's what has increased coverage. (P11)*

This health system transformation has also fostered a collaborative approach between the public and private sectors, creating multiple pathways for people to access HIV and viral hepatitis testing services. While public facilities successfully implemented free basic testing programmes, they recognised limitations in their diagnostic capacity, particularly for specialised tests such as quantitative PCR for hepatitis D. In response, health facilities developed innovative solutions through formal partnerships with private laboratories, ensuring that patients could access a full range of necessary diagnostic services at reduced costs:

> *Our director has even contracted [private] laboratory to do it... If earlier these tests were very expensive, it was unaffordable for the patient. And now the hospital has a contract, we refer them for hepatitis D testing... And I personally refer them there, there is a 50% discount. (P7)*

Finally, the national hepatitis B vaccination programme has created additional opportunities to expand testing coverage in primary care settings. By offering vaccination to everyone, the programme has helped to normalise hepatitis testing as part of routine preventive care and reached people who might not otherwise seek testing. As one doctor explained:

> *...we have been testing since 2023, we have a program on hepatitis B [...], especially on hepatitis B immunisation. For example, absolutely healthy people, we invited them, they came to us, we had a separate room for testing [...] after 20 minutes the result was negative, then we took them to the immunisation room. (P14)*

**Motivation: Professional responsibility as a driving force.** Doctors in PHC expressed a strong sense of professional responsibility that acted as a key motivator for providing testing services. This commitment manifested itself in a stronger awareness of their role in both individual patient care and wider public health surveillance. Doctors in PHC recognised their obligation to maintain ongoing relationships with patients and ensure continuity of care, as one doctor pointed out:

> *I have said I tell them to come back to us after some time, it's our responsibility, we have to monitor the patients for life. (P3)*

We found that doctors saw themselves as guardians of public health, emphasising the need for vigilance and proactive approaches to testing. This professional commitment was reflected in their approach to testing decisions and their understanding of their broader societal role:

> *First of all, perhaps doctors should be more responsible. We should be vigilant. (P2)*

This sense of professional duty acted as a counterbalance to the perceived lack of patient responsibility described above, with PHC doctors seeing their own accountability as crucial to effective testing. At the same time, doctors demonstrated strong willingness to learn more about HIV and viral hepatitis testing, mentioning that these trainings not only enhance their knowledge about these infections, but also help to develop some soft skills to communicate with patients.

## Part II: quantitative results

This section presents the findings from our quantitative analysis of barriers and facilitators to HIV and viral hepatitis testing in PHC in the Kyrgyz Republic. In total, 1,096 doctors in PHC completed the online questionnaire. We excluded 16 respondents due to reporting working outside PHC. This led to a final dataset of 1,080 doctors in PHC who were included in our analyses.

The majority of the participants were female (89%), and most respondents were in the age groups 51–60 years (27%) and 31–40 years (23%) followed by the age group of 61+ (20%). When asked about hepatitis B vaccination, 85% of doctors participating in the survey responded ever being vaccinated, of which 82% reported being fully vaccinated (at least three doses). Detailed sociodemographic and healthcare practice characteristics are reported in Table 1.

**Table 1. Distribution of socio-demographic and PHC facility characteristics of the study population (n = 1,080).**

| | Characteristics | Number | Proportion (%) |
|---|---|---|---|
| **Gender** | Female | 953 | 89 |
| | Male | 113 | 11 |
| | Missing | 14 | -- |
| **Age group (years)** | 30 or younger | 141 | 13 |
| | 31-40 | 241 | 23 |
| | 41-50 | 165 | 16 |
| | 51-60 | 282 | 27 |
| | 61+ | 215 | 20 |
| | Do not want to answer | 9 | 0.8 |
| | Missing | 27 | -- |
| **Region (of PHC facility)** | Batken | 48 | 4.5 |
| | Bishkek | 320 | 30 |
| | Chui | 111 | 10 |
| | Issyk-Kule | 135 | 13 |
| | Manas | 235 | 22 |
| | Naryn | 23 | 2.2 |
| | Osh Region | 38 | 3.6 |
| | Talas | 131 | 12 |
| | Osh City | 24 | 2.3 |
| | Missing | 15 | -- |
| **Geographical area (of PHC facility) (North/South)** | North | 720 | 68 |
| | South | 345 | 32 |
| | Missing | 15 | -- |
| **Urbanization level (of primary healthcare practice)** | Urban | 571 | 54 |
| | Rural | 491 | 46 |
| | Missing | 18 | -- |
| **Healthcare specialty** | Family medicine | 743 | 70 |
| | Infectious Diseases | 47 | 4.4 |
| | Dermatovenerology | 12 | 1.1 |
| | Internal Medicine | 76 | 7.1 |
| | Obstetrics and Gynaecology | 98 | 9.2 |
| | Other | 151 | 12 |
| **Ever vaccinated against hepatitis B** | Yes | 775 | 85 |
| | No | 124 | 14 |
| | Do not know | 13 | 1.4 |
| | Missing | 168 | -- |
| **Fully vaccinated against hepatitis B (at least three doses) (n = 775)** | Yes, at least three doses | 632 | 82 |
| | No, only one or two doses | 102 | 13 |
| | I do not know | 38 | 4.9 |
| | Missing | 3 | -- |

## Testing activities

Of the 1,080 participants, 892 (91%) reported that free hepatitis B tests were available in their facility, 641 (65%) had free hepatitis C tests and 749 (76%) had free HIV tests. In total, 845 (80%), 633 (60%) and 704 (67%) reported having performed HBV, HCV or HIV testing in the past 12 months.

Of those who had performed testing of either HBV, HCV or HIV in the past 12 months (n = 904), 739 participants (82%) indicated that they had also performed integrated testing of patients. Of those who had not performed integrated testing (n = 160), the most common reasons for not offering were that patients presented with viral hepatitis symptoms and therefore HIV test was not offered (n = 37 (23%)), that tests for integrated testing were not available (22 (14%)) and 19 participants (12%) indicated that they did not think integrated testing was a good approach. Knowledge was an issue for 14 participants (8.8%) who indicated not knowing what integrated testing was, and 8 (5.0%) who reported not being able to explain integrated testing to patients. Of the overall 1,080 participants, 134 (13%) reported not to have tested patients despite presenting with clinical indications the last 12 months.

### Barriers and facilitators to HIV and viral hepatitis testing

Overall, the barriers for HBV, HCV and/or HIV testing, to which most participants agreed or strongly agreed were the need of more viral hepatitis (66%) and HIV (62%) training, lack of free testing and treatment for HDV (60%) and that patients decline hepatitis (24%) and HIV (26%) testing when offered in the healthcare facility (Fig 1).

The facilitators for HBV, HCV and/or HIV testing to which most participants agreed or strongly agreed were access to free testing and treatment for hepatitis B and C (92%) and HIV (89%) makes it easier to offer testing, and their knowledge whom to test for HBV and HCV (79%) and HIV (83%) (Fig 2).

When stratifying the three most frequently reported barriers (defined by responses of agree or strongly agree) by geographical location no significant differences were observed between doctors in PHC working in rural versus urban settings, nor between those in the Northern versus Southern regions (of the Kyrgyz Republic. Similarly, stratification by age group revealed comparable proportions across categories, with no statistically significant differences identified (all $p > .05$) [see S4 Appendix].

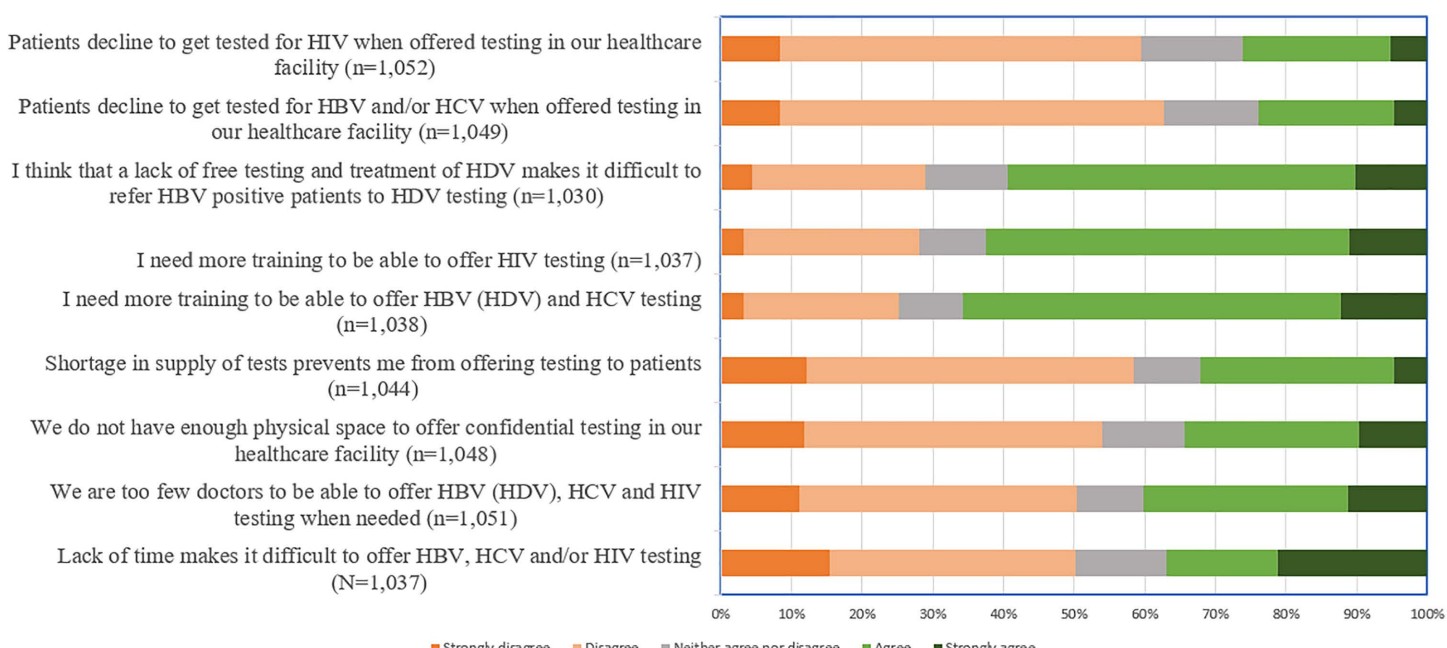

**Fig 1. Barriers for HBV, HCV and HIV testing (n = 1,080).**

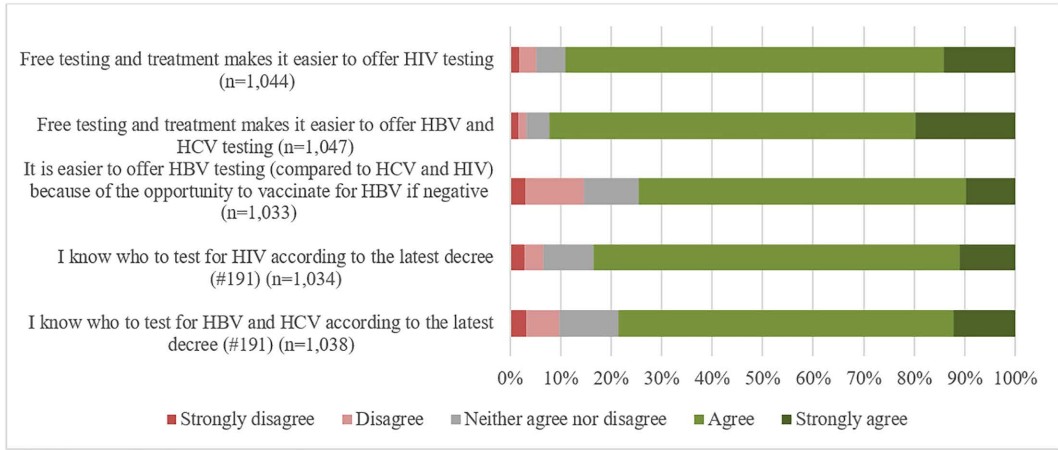

**Fig 2. Facilitators for HBV, HCV and HIV testing (n = 1,080).**

## Key populations

In total, 781 participants (77%) reported not seeing any members of key populations as patients in their PHC facility. Overall, 159 (16%) of participants reported seeing people who inject drugs in their practice, 107 (11%) reported seeing sex workers, and 96 (9.0%) reported seeing people in prison and other closed settings. In total, 92 (9.0%) and 62 (6.0%) participants reported seeing men who have sex with men and trans and gender diverse people in their practice, respectively.

Participants were asked, if given the choice, if they would prefer not to provide care or services to key population groups. People who inject drugs and men who have sex with men were the groups to which most doctors responded that they would prefer not to provide services for (Fig 3).

The participants who agreed to not wanting to offer healthcare services to key populations were asked what the main reasons were. Across all key populations, lack of training was frequently reported, while the other reasons differed according to key population group (Table 2).

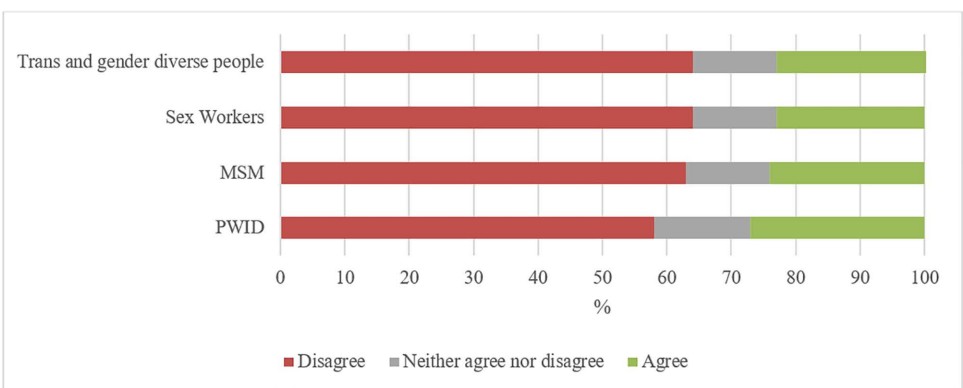

**Fig 3. Would prefer to not to offer healthcare service to key populations (n = 1,080).**

**Table 2. Reasons for not wanting to provide services to key populations.**

| Key population | They put me at higher risk for disease n (%) | This group engages in immoral behaviour n (%) | I have not received training to work with this group n (%) |
|---|---|---|---|
| People who inject drugs (n = 273) | 65 (75) | 97 (38) | 120 (47) |
| Men who have sex with men (n = 223) | 40 (19) | 111 (51) | 94 (44) |
| Sex workers (n = 193) | 44 (23) | 85 (45) | 91 (48) |
| Trans and gender diverse people (n = 182) | 35 (20) | 87 (49) | 85 (48) |

## Discussion

Our study highlights that for doctors in PHC offering HIV and viral hepatitis testing is still complex, and prominent barriers exist preventing successful implementation of testing on PHC level in the Kyrgyz Republic. Particularly lack of training, especially on risk factors and key populations, was reported both through the IDIs and the questionnaire. Our study also identified important facilitators that encourage testing in PHC such as access to free testing and treatment and knowledge of who to offer testing.

Nonetheless, our study population reported overall good testing practices. More than 80% informed performing integrated testing for HBV, HCV and HIV, and only 13% of questionnaire participants stated that sometimes patients were not tested despite presence of clinical indication. This was also found in the IDIs, during which some participants expressed that they felt capable and able to offer testing due to their detailed knowledge of the risks of transmission and long-term consequences of undiagnosed and untreated infections.

Through the questionnaire, we found that a lack of training was one of the main barriers for offering testing, and one of the main facilitators were knowledge of who to test according to the MoH guidelines. This was further elaborated on in the IDIs, where doctors mentioned that they felt knowledgeable about clinical indications for testing, but were missing more information about epidemiological criteria as well as how to accurately assess levels of risk. Lack of training and need for continuing education reflect what was also found in other studies [29–31].

Risk-based testing informed by exposure to risks remains a challenge in PHC, which is also found in other studies [32,33]. It often warrants a different testing strategy in PHC to improve provider-initiated testing such as indicator-condition guided testing [34] or screening during routine check-ups [16]. In our study, a large proportion of participating doctors working in PHC reported performing integrated HIV and viral hepatitis testing. This is positive, and given that viral hepatitis is more generalised and seems less stigmatised, it provides a way to by-pass stigma and discrimination associated with risk factors and HIV and could contribute to normalising HIV testing by offering integrated testing for viral hepatitis and HIV in PHC. Furthermore, at a time of a changing global health landscape with decreased funding, disruption of services and staff availability caused by, e.g., funding cuts by the government of the United States, integrated testing and people-centred rather than disease-centred delivery models are imperative to secure efficient use of resources and structures [1,35]. However, lack of resources and availability of test supplies was mentioned in the IDIs, and it is key that resources also for viral hepatitis testing and treatment are made available in the country.

Other studies have shown that barriers to HIV and viral hepatitis testing are often more prevalent among older doctors and in more rural and/or traditional areas [36]. While doctors in PHC expressed different views depending on whether working in the Northern or Southern part of the country, this was not confirmed in our quantitative analysis. The distribution of barriers (and facilitators) was similar when stratified by age and geographical region. This may be because the area of HIV and viral hepatitis testing in PHC is still relatively new and not yet so developed for geographical and age-related differences to crystalize. Once doctors in PHC get more exposed to testing and sub-populations at higher risk of infection, these differences may become more visible and found through later studies on this topic from the Kyrgyz Republic. It may also be that these barriers and facilitators are so common and generalized that they apply across all sectors, population groups and geographical areas in the Kyrgyz Republic.

 

During the interviews, stigma and discrimination came up as a recurrent theme and a barrier for testing. Given that key populations are most affected by HIV, ensuring access to testing for this group is central in the HIV and viral hepatitis response. The majority of doctors reported not seeing key populations as patients, which indicates existing barriers on social, legal and structural levels, and these may become even worse in the coming years as a consequence of the law specifically targeting lesbian, gay, bisexual, transgender, and queer or questioning (LGBTQI+) communities [37]. There are probably different explanations for why doctors report not seeing key populations. It might be due to key populations not disclosing this information due to fear of stigma or discrimination, that they are not asked by their doctor, or if asked they might fear lack of confidentiality. It may also be that they are either tested elsewhere or not at all. In a survey conducted in the European Union (EU)/European Economic Area (EEA) countries, one in four participants reported having been worried about being treated differently by healthcare workers due to their HIV status [38]. Studies have also found that stigma and discrimination prevent key populations from accessing healthcare and worsen and delay testing and access to care [26,39].

The latest report on stigma in the Kyrgyz Republic among 708 people living with HIV found that internal stigma played a large role in overall quality of life and disclosing of HIV status But also that external stigma from external environments including healthcare workers played a role with, e.g., 31% delaying seeking care because of being scared that healthcare workers would treat them badly or disclose status without their consent [40]. In our study, a noteworthy proportion of doctors reported preferring not to provide care for key populations (17–25%), and expressed that rise in infections was caused by social and immoral behaviour. Most important reasons for not wanting to provide care were a lack of training and attributed immoral behaviour to key populations by the doctors, in particular for addressing the needs of men who have sex with men. This requires work on a system, provider and patient level through information and education, which is ongoing in the country, but at the same time challenged by the new law prohibiting the dissemination of information about LGBTQI+ [37].

Improving testing is a public health challenge, with strong behavioural components both on patient (population) and provider level. In addition to understanding reported barriers and facilitators by doctors in PHC, it is equally important to understand the social and cultural context, which drive doctors to think and act as they do. The application of the COM-B framework to understand the context in which the barriers and facilitators are experienced by the doctors is important to comprehend the components needed for behaviour change and thereby being able to develop targeted interventions that are socially and culturally acceptable.

The most important strength of this study was the combination of qualitative and quantitative methods. The data from the IDIs provided contextual information and understanding of barriers and facilitators from the perspective of doctors in PHC in the Kyrgyz Republic. This enabled us to ensure that the questions and answer categories included in the questionnaire were relevant to the study population and our objectives. Additionally, during IDIs, participants could express their opinions and views on especially barriers and challenges in more detail, and prompts by the interviewers allowed a deeper and more nuanced understanding of underlying messages and opinions. Moreover, observations of non-verbal communication permitted also studying bodily and facial expressions when talking about certain topics. This was particularly important for sensitive topics such as risk factors. In an online questionnaire, with pre-defined answering categories completed anonymously and independently by the doctors in PHC, this was not possible. Another strength for the online questionnaire is that more than double the target sample size was reached. While lack of recent figures on doctors working in PHC and details on sociodemographic characteristics prevent accurately evaluating representativeness, the distribution in terms of age, sex and geographical location mirrors the distribution of doctors and nurses based on data from WHO with more doctors and nurses in Urban areas, more female doctors and more than 50% of family doctors at retirement age [8]. Moreover, the large number of responses have likely reduced risk of bias and underrepresentation of some groups.

Our study is also subject to important limitations. First of all, there was a risk of selection bias, and only doctors in PHC who had capacity and time for participation or who wanted to express their opinions took part in our study. Moreover,

despite anonymity, it may be that participants felt uncomfortable providing negative information, or information which is deemed socially unacceptable, in particular given the involvement of the MoH, and therefore answers may reflect a more positive view than is actually the case. As the online questionnaire was distributed through several channels through an open non-personalised link, it may be that our participants include people not fulfilling our inclusion criteria.

To our knowledge, our study is the first to outline barriers and facilitators for HIV and viral hepatitis testing in PHC in the Kyrgyz Republic. While the PHC level has the potential to accelerate progress towards reaching the goals set by WHO on testing, it is essential that sufficient knowledge, capacity and resources are available to doctors in PHC, including increasing the number of well-trained PHC doctors in rural areas of the country. By identifying existing barriers and facilitators in the study, tailored interventions targeting training, knowledge enhancement for HIV and viral hepatitis, including risk factors and key populations, which can enable testing increase in PHC, can be developed. Post-graduate training for doctors in PHC is an important intervention to address knowledge gaps preventing testing, and is consistent with our finding that doctors in PHC want more training and knowledge to be better prepared for their task of testing for viral hepatitis and HIV.

Interventions, including training, need to be appropriate according to social and cultural norms that stipulate whether or not a certain action and behaviour is acceptable. The impact of barriers and facilitators on system and patient level are equally important to understand the overall systemic context in which barriers and facilitators for testing exist and interplay such as lack of HCV tests. To improve testing, all levels need to be understood and targeted with relevant interventions, such as lack of access to rapid tests This was however beyond the scope of our study, but should be covered in future research. Moreover, it would be helpful for future research to look at the impact of barriers and facilitators on the testing offer and uptake in PHC, and use baseline numbers to look at the impact of any implemented interventions, for example training for doctors in PHC.

Targeted interventions, possibly by building on the COM-B framework, and inclusion of the doctors working in PHC through a participatory approach will help doctors offer and conduct HIV and viral hepatitis testing to all population groups in need of testing in the Kyrgyz Republic.

## Supporting information

**S1 Appendix. Interview guide.**
(DOCX)

**S2 Appendix. Questionnaire.**
(DOCX)

**S3 Appendix. Coding scheme.**
(PDF)

**S4 Appendix. Stratified analyses.**
(DOCX)

**S1 File. Inclusivity in global research.**
(DOCX)

## Acknowledgments

First of all, we thank all the doctors in PHC who participated for their time and contribution to the study. Further acknowledgements go to colleagues from the WHO Regional Office for Europe and the WHO country office in the Kyrgyz Republic for their input and logistical support. We would also like to thank Rishat Azikhanov and Saikal Kylychbekova for piloting our interview guide and providing important contextual information for the study.

## Author contributions

**Conceptualization:** Ida Sperle, Nikolay Lunchenkov, Anastassiya Stepanovich-Falke, Michael Brandl, Olena Kysil, Barbara Gunsenheimer-Bartmeyer, Sandra Dudareva.

**Data curation:** Nikolay Lunchenkov, Anastassiya Stepanovich-Falke, Aisuluu Bolotbaeva.

**Formal analysis:** Ida Sperle, Nikolay Lunchenkov.

**Methodology:** Ida Sperle, Nikolay Lunchenkov.

**Project administration:** Olena Kysil.

**Supervision:** Zuridin S. Nurmatov, Aybek A. Bekbolotov, Barbara Gunsenheimer-Bartmeyer, Sandra Dudareva.

**Validation:** Ida Sperle.

**Writing – original draft:** Ida Sperle, Nikolay Lunchenkov.

**Writing – review & editing:** Ida Sperle, Nikolay Lunchenkov, Anastassiya Stepanovich-Falke, Zuridin S. Nurmatov, Aybek A. Bekbolotov, Aisuluu Bolotbaeva, Aman Taalaibekov, Zamira Abdrakhmanova, Zulpueva Akmaral Aidarbekovna, Michael Brandl, Olena Kysil, Stela Bivol, Viviane Bremer, Barbara Gunsenheimer-Bartmeyer, Sandra Dudareva.

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
