## [Decision Letter · Decision Letter 0]

31 Jul 2025

Dear Dr. Sperle,

Thank you for submitting your manuscript to PLOS ONE. After careful consideration, we feel that it has merit but does not fully meet PLOS ONE’s publication criteria as it currently stands. Therefore, we invite you to submit a revised version of the manuscript that addresses the points raised during the review process.

We look forward to receiving your revised manuscript.

Kind regards,

Jason T. Blackard, PhD

Academic Editor

PLOS ONE

3. In the online submission form, you indicated that [Data cannot be shared publicly because of because of data protection. Data are available from the corresponding author upon reasonable request.].

Additional Editor Comments:

This is a cross-sectional, mixed methods study of facilitators and barriers to HIV and viral hepatitis in the Kyrgyz Republic.

The study rationale and methods utilized are well described and appropriate.

It is unclear who the HCWs are?  Doctors?  Nurses?  Medical students?  Who identified the HCWs that would be told about the online survey?  How representative are the HCW survey respondents of the entire HCW population in the Kyrgyz Republic?

Are there substantive differences across different regions of the country?

Are HIV, HBV, and HCV testing and treatment offered free of charge in every region of the country?

Reviewers' comments:

Reviewer's Responses to Questions

**Comments to the Author**

1. Is the manuscript technically sound, and do the data support the conclusions?

Reviewer #1: Yes

Reviewer #2: Partly

2. Has the statistical analysis been performed appropriately and rigorously?

Reviewer #1: Yes

Reviewer #2: No

3. Have the authors made all data underlying the findings in their manuscript fully available?

Reviewer #1: No

Reviewer #2: Yes

4. Is the manuscript presented in an intelligible fashion and written in standard English?

Reviewer #1: Yes

Reviewer #2: Yes

Reviewer #1: As many other WHO supported countries Kirgiz Republic underwent a significant reform in HIV care delivery, namely decentralization of HIV and hepatitis care. It is therefore of great importance and interest to review barriers and facilitators for HIV and hepatitis testing in this new settings. The paper shows strong methodological construct of work performed and delivers important message directly from HCWs. Therefore I strongly support its publication with some minor additions listed below:

General comments:

1. Please revise the language throughout the paper to meet “People first language” consensus (eg. Do not use abbreviations like PLHIV or PWID).

2. Please provide details for the approval by the ethics committee of the institutional review board in the Kyrgyz Republic, which is the name of the board, location and the decision number.

3. Please provide in introduction a short description of the structure of care for HIV and hepatitis in KR, including the information that after decentralization each primary care center has infectiologist who is responsible for HIV and hepatitis patients. At the same time family doctors received new task of HIV and hepatitis testing. There are also several systematic barriers here, such as lack of access to rapid tests (except for pregnant women) and responsibility of delivering epidemiological investigation for the epidemiology (sometimes performed by epidemiologist depending of particular settings).

4. An important finding is that HCWs are interested in receiving more professional postgraduate training in additional to internal training provided under auspices of MoH. Could you reflect more on that, basing on the qualitative interviews? This could provide a guidance for international societies on the need and guide their actions.

Reviewer #2: This manuscript reports on a mixed qualitative and quantitative examination of primary healthcare workers’ perceived barriers and facilitators of HIV and viral hepatitis testing in the Kryrgyz Republic. They report on qualitative interviews conducted with 21 primary healthcare providers and quantitative survey data from 1080 providers. Descriptive statistics were used to characterize HIV and viral hepatitis testing barriers and facilitators. Key facilitators of testing included access to free testing, knowledge regarding indicators for testing, and ability to provide linkage to vaccination; the core barrier was lack of experience working with this patient population.

The following are suggestions for the authors to improve the manuscript.

-The introduction is well-written. It would be helpful to provide some additional information to characterize the nature of primary healthcare within the Kyrgyz Republic as some readers may not be familiar with the services provided, the extent to which healthcare is covered (e.g., cost, insurance, etc.), and the extent to which the general population and specialized populations noted (e.g., individuals who inject drugs) utilize these services. Additionally, national (or local) guidelines for screening (e.g., risk screening questionnaires within primary healthcare) or testing would be helpful.

-I would have liked to know a bit more about the population of healthcare workers sampled for both the qualitative and quantitative portions of the study. For example, what professions were included?

-The qualitative analytic approach is well described and rigorous. Could the authors provide additional rationale for the use of the COM-B framework employed? The introduction and methods did not indicate a particular framework would be utilized, so it would be helpful to provide some additional rationale.

-I would have liked to see a bit more robust approach to the quantitative analyses. While presenting the descriptive information is useful, I found myself wondering why additional analyses weren’t considered (e.g., grouping all clinics into Northern vs Southern, dichotomizing age into 18-50 and 51+). I found the qualitative methods and presentation of results to be rigorous, but the quantitative portion was lacking in the similar level of rigor.

-For the quantitative survey, the results could be more clearly presented. For example, when reporting on vaccination status, was this for the participants or for patients in their clinics?

-I would like to have seen a bit more information about future research directions. For example, any consideration of examining the extent to which identified barriers/facilitators map onto actual clinical practice (e.g., prevalence of testing in clinics)?

**Do you want your identity to be public for this peer review?** For information about this choice, including consent withdrawal, please see our Privacy Policy

Reviewer #1: **Yes: ** Justyna Kowalska, Medical University of Warsaw, Poland

Reviewer #2: No

---

## [Author Response · Author response to Decision Letter 1]

26 Sep 2025

Additional Editor Comments:

This is a cross-sectional, mixed methods study of facilitators and barriers to HIV and viral hepatitis in the Kyrgyz Republic.

The study rationale and methods utilized are well described and appropriate.

It is unclear who the HCWs are? Doctors? Nurses? Medical students? Who identified the HCWs that would be told about the online survey? How representative are the HCW survey respondents of the entire HCW population in the Kyrgyz Republic?

Author response: Thank you for this comment and questions. We included medical doctors who worked in primary healthcare. We’ve added more details on this in the abstract (L27) as well as in the introduction (L126-127) and methods section (L140). Based on this comment as well as similar comments from the other reviewer, we have re-written this, so instead of referring to HCWs we refer to doctors working in PHC.

The doctors were invited with help from the research team in the Kyrgyz Republic who coordinated in-country activities. This included the organisation of the IDIs and the distribution of the online questionnaire through regional health coordinators, who then contacted the PHC facilities. The doctors invited were thereby identified via the regional health coordinators in the country.

Regarding representativeness, the large number of responses is positive and likely reduces the risk of bias. Although we don’t have recent numbers on sociodemographic characteristics of the target population, the distribution in terms of age, sex and geography of our study population seems to mirror that of the target population with a larger proportion of females and older age-groups. A few thoughts on this have also been added to the discussion in L735-740.

Are there substantive differences across different regions of the country?

Author response: Yes, there is an unequal distribution of healthcare workers and thus doctors in PHC in the country and large differences across regions. There are fewer doctors, also PHC doctors in rural areas of the country compared to urban areas and especially the capital Bishkek, and there is a problem with aging population of doctors, also PHC doctors, which is a larger problem in rural areas We’ve added some sentences on this in the discussion (L 735-740).

Are HIV, HBV, and HCV testing and treatment offered free of charge in every region of the country?

Author response: Yes, this is provided free of charge also through primary health care and regional AIDS Centres (state funded). For viral hepatitis, testing and treatment has been made available free of charge since 2023. However, a difference in PHC capacity in rural areas versus urban probably causes skewed access. We’ve added more on this in the introduction (L 88-98).

Reviewer #1: As many other WHO supported countries Kirgiz Republic underwent a significant reform in HIV care delivery, namely decentralization of HIV and hepatitis care. It is therefore of great importance and interest to review barriers and facilitators for HIV and hepatitis testing in this new settings. The paper shows strong methodological construct of work performed and delivers important message directly from HCWs. Therefore I strongly support its publication with some minor additions listed below:

Author response: Many thanks for reading our manuscript and for the valuable comments and suggestions for improvement. Detailed responses, and how and where changes are implemented are stated below.

General comments:

1. Please revise the language throughout the paper to meet “People first language” consensus (eg. Do not use abbreviations like PLHIV or PWID).

Author response: thank you for raising this, abbreviations including PLHIV, PWID and MSM have been spelled out throughout the text.

2. Please provide details for the approval by the ethics committee of the institutional review board in the Kyrgyz Republic, which is the name of the board, location and the decision number.

Author response: More details have been added in L294-296.

3. Please provide in introduction a short description of the structure of care for HIV and hepatitis in KR, including the information that after decentralization each primary care center has infectiologist who is responsible for HIV and hepatitis patients. At the same time family doctors received new task of HIV and hepatitis testing. There are also several systematic barriers here, such as lack of access to rapid tests (except for pregnant women) and responsibility of delivering epidemiological investigation for the epidemiology (sometimes performed by epidemiologist depending of particular settings).

Author response: Thanks for this good point, more information on the structure of care in PHC has been added to the introduction (L97-98), and systemic barriers mentioned in the discussion (L762-765).

4. An important finding is that HCWs are interested in receiving more professional postgraduate training in additional to internal training provided under auspices of MoH. Could you reflect more on that, basing on the qualitative interviews? This could provide a guidance for international societies on the need and guide their actions.

Author response: yes, this was an interesting finding and important for future action in this area. We’ve added a few more thoughts on this in the discussion (L 756-759).

Reviewer #2: This manuscript reports on a mixed qualitative and quantitative examination of primary healthcare workers’ perceived barriers and facilitators of HIV and viral hepatitis testing in the Kryrgyz Republic. They report on qualitative interviews conducted with 21 primary healthcare providers and quantitative survey data from 1080 providers. Descriptive statistics were used to characterize HIV and viral hepatitis testing barriers and facilitators. Key facilitators of testing included access to free testing, knowledge regarding indicators for testing, and ability to provide linkage to vaccination; the core barrier was lack of experience working with this patient population.

Author response: Thank you for reading our manuscript and providing comments and suggestions on how to improve our manuscript. Detailed responses, and how and where changes are implemented are written below.

The following are suggestions for the authors to improve the manuscript.

-The introduction is well-written. It would be helpful to provide some additional information to characterize the nature of primary healthcare within the Kyrgyz Republic as some readers may not be familiar with the services provided, the extent to which healthcare is covered (e.g., cost, insurance, etc.), and the extent to which the general population and specialized populations noted (e.g., individuals who inject drugs) utilize these services. Additionally, national (or local) guidelines for screening (e.g., risk screening questionnaires within primary healthcare) or testing would be helpful.

Author response: Thank you, we have added more information also on the general healthcare and insurance system in the introduction (L78-83, L88-98) and point on key populations (L118-121).

-I would have liked to know a bit more about the population of healthcare workers sampled for both the qualitative and quantitative portions of the study. For example, what professions were included?

Author response: We included medical doctors working in primary healthcare settings. We’ve added more details on this in the abstract (L27) as well as in the introduction (L126-127) and methods section (L140). We have also written this explicitly instead of using the HCW abbreviation throughout the manuscript.

-The qualitative analytic approach is well described and rigorous. Could the authors provide additional rationale for the use of the COM-B framework employed? The introduction and methods did not indicate a particular framework would be utilized, so it would be helpful to provide some additional rationale.

Author response: Thanks, we’ve added one sentence about the COM-B framework in the introduction and methods mentioning COM-B (l129-130 & 255-257).

-I would have liked to see a bit more robust approach to the quantitative analyses. While presenting the descriptive information is useful, I found myself wondering why additional analyses weren’t considered (e.g., grouping all clinics into Northern vs Southern, dichotomizing age into 18-50 and 51+). I found the qualitative methods and presentation of results to be rigorous, but the quantitative portion was lacking in the similar level of rigor.

Author response: Thanks for this good point. For the quantitative analyses, we unfortunately do not know which clinics answered, only participants and then in which geographical location their PHC facility is located. We grouped the responding doctors into “south” and “north” and also by age (18-50 years and 51-61+), and did additional stratified analyses using Pearson’s Chi-squared to look for significant differences with the most frequent barriers. We did not find any significant differences, and therefore decided to include these results as appendix to the manuscript but we have added p-value in the text (L626) (Appendix 3).

We have now also added some more results to the text in the manuscript, and a few additional quantitative results as well to better match the qualitative part of the manuscript (L 601-607, 611-612, 616-617).

-For the quantitative survey, the results could be more clearly presented. For example, when reporting on vaccination status, was this for the participants or for patients in their clinics?

Author response: this was self-reported vaccination status among participants in the questionnaire. We’ve added clarified text in Table 1, and lines 227-228 and 590-591.

-I would like to have seen a bit more information about future research directions. For example, any consideration of examining the extent to which identified barriers/facilitators map onto actual clinical practice (e.g., prevalence of testing in clinics)?

Author response: good point thanks, a few ideas have been added in the discussion, L766-768.

---

## [Decision Letter · Decision Letter 1]

23 Oct 2025

Barriers and Facilitators to HIV and Viral Hepatitis Testing in Primary Healthcare Settings in the Kyrgyz Republic: A Mixed-Methods Study Using the COM-B Framework

PONE-D-25-34739R1

Dear Dr. Sperle,

We’re pleased to inform you that your manuscript has been judged scientifically suitable for publication and will be formally accepted for publication once it meets all outstanding technical requirements.

Kind regards,

Jason T. Blackard, PhD

Academic Editor

PLOS ONE

Additional Editor Comments (optional):

None

Reviewers' comments:

Reviewer's Responses to Questions

**Comments to the Author**

Reviewer #2: All comments have been addressed

2. Is the manuscript technically sound, and do the data support the conclusions?

Reviewer #2: Yes

3. Has the statistical analysis been performed appropriately and rigorously?

Reviewer #2: Yes

4. Have the authors made all data underlying the findings in their manuscript fully available?

Reviewer #2: No

5. Is the manuscript presented in an intelligible fashion and written in standard English?

Reviewer #2: Yes

Reviewer #2: (No Response)

**Do you want your identity to be public for this peer review?** For information about this choice, including consent withdrawal, please see our Privacy Policy

Reviewer #2: No

---

## [Editor Report · Acceptance letter]

PONE-D-25-34739R1

PLOS ONE

Dear Dr. Sperle,

I'm pleased to inform you that your manuscript has been deemed suitable for publication in PLOS ONE. Congratulations! Your manuscript is now being handed over to our production team.

Kind regards,

on behalf of

Dr. Jason T. Blackard

Academic Editor

PLOS ONE